# Marine Natural Products and Drug Resistance in Latent Tuberculosis

**DOI:** 10.3390/md17100549

**Published:** 2019-09-26

**Authors:** Muhammad Tahir Khan, Aman Chandra Kaushik, Aamer Iqbal Bhatti, Yu-Juan Zhang, Shulin Zhang, Amie Jinghua Wei, Shaukat Iqbal Malik, Dong Qing Wei

**Affiliations:** 1Department of Bioinformatics and Biosciences, Capital University of Science and Technology, Islamabad 44000, Pakistan; tahirmicrobiologist@gmail.com (M.T.K.); drshaukat@cust.edu.pk (S.I.M.); 2The State Key Laboratory of Microbial Metabolism, College of Life Sciences and Biotechnology, Shanghai Jiao Tong University, Shanghai 200240, China; amanbioinfo@gmail.com; 3Department of Electrical Engineering, Capital University of Science and Technology, Islamabad 44000, Pakistan; aib@cust.edu.pk; 4College of Life Sciences, Chongqing Normal University, Chongqing 401331, China; zhangyj@cqnu.edu.cn; 5Department of Immunology and Microbiology, School of Medicine, Shanghai Jiao Tong University, Shanghai 200025, China; shulinzhang@sjtu.edu.cn (S.Z.);

**Keywords:** marine anti-TB compounds, PZA, MTB, latent TB, sponges

## Abstract

Pyrazinamide (PZA) is the only drug for the elimination of latent *Mycobacterium tuberculosis* (MTB) isolates. However, due to the increased number of PZA-resistance, the chances of the success of global TB elimination seems to be more prolonged. Recently, marine natural products (MNPs) as an anti-TB agent have received much attention, where some compounds extracted from marine sponge, Haliclona sp. exhibited strong activity under aerobic and hypoxic conditions. In this study, we screened articles from 1994 to 2019 related to marine natural products (MNPs) active against latent MTB isolates. The literature was also mined for the major regulators to map them in the form of a pathway under the dormant stage. Five compounds were found to be more suitable that may be applied as an alternative to PZA for the better management of resistance under latent stage. However, the mechanism of actions behind these compounds is largely unknown. Here, we also applied synthetic biology to analyze the major regulatory pathway under latent TB that might be used for the screening of selective inhibitors among marine natural products (MNPs). We identified key regulators of MTB under latent TB through extensive literature mining and mapped them in the form of regulatory pathway, where SigH is negatively regulated by RshA. PknB, RshA, SigH, and RNA polymerase (RNA-pol) are the major regulators involved in MTB survival under latent stage. Further studies are needed to screen MNPs active against the main regulators of dormant MTB isolates. To reduce the PZA resistance burden, understanding the regulatory pathways may help in selective targets of MNPs from marine natural sources.

## 1. Introduction

The latent state of tuberculosis (TB) is asymptomatic, but poses a risk in developing the active state of TB during the lifetime. According to the latest World Health Organization (WHO) report in 2018, TB is the leading public health problem among infectious diseases resulting from a single infectious agent, ahead of HIV/AIDS, and is the ninth leading cause of death worldwide. Approximately 1.3 million TB deaths occurred in 2017 excluding 374,000 deaths (10%) among HIV-positive individuals among 10.4 million total TB incidents (90% adults). About 1.7 billion people (23% of the world’s population) are estimated to have a latent TB infection, indicating a risk of developing active TB during their lifetime. India, Indonesia, China, Philippines, and Pakistan are the top five countries comprising 56% of the world’s estimated TB cases [1]. Among the infected individuals, 5–10% develop active TB. Such individuals suffer from latent TB, where the *Mycobacterium tuberculosis* (MTB) resides in alveolar macrophages in a non-replicative form (latent TB) [2,3,4]. The risk of developing active TB from non-replicative forms has been accounted in 10% of cases in latently infected populations [2,3,5], but may increase in cases of TB-HIV co-infections, immunosuppressive therapy, and old age [6,7,8,9,10,11]. Recently, a large number of studies reported drug resistances in TB [12,13,14] effecting the global TB control program. 

### 1.1. PZA against Latent TB

Among the available anti-tuberculosis agents, pyrazinamide (PZA) is the only drug that is active against non-replicative MTB [15,16,17,18]. The host generates different types of stresses to eliminate the MTB isolates effectively. However, the organism switches a sensory system that generates a complex signaling network, assisting in entry into the latent state [19,19,20,21,22]. Before conversion into the latent stage, MTB faces a number of oxidoreductive stress in alveolar macrophages of the host including oxidative, acidic, and nitrative stress. These stresses are vital in the transition from active (replicative) TB into latent (non-replicative) state [23,24]. 

### 1.2. Signaling in Latent TB

The genome of MTB strains have diverse stress responders, switching on the genetic program for transition into latency [25,26]. Among these sensors under the latent stage are sigma (s) factors, which are the primary regulators of gene expression. MTB genomes encoded 13 factors of the sigma 70 family [27], which are categorized into four groups known as S1, 2, and 3 including SigA, SigB, and SigC, respectively, while the remaining one belongs to group 4, mainly involved in extra-cytoplasmic sensing and signaling [28,29,30]. These regulators have been called “S” factors due to their role in growth and stress conditions [28]. MTB senses redox through SigH, SigE, SigF, and SigL encoded regulators, playing a critical role in survival under extreme conditions [23,30,31]. Fernandes et al. first demonstrated that the role of SigH in oxidative stress [29] was also involved in the expression of thioredoxins (trxB1 and TrxC) and thioredoxin reductase, while the stress-responsive “S” factor and SigE helped mitigate oxidative stress. The “S” factor, along with SigB expression, is also regulated by SigE and SigH. [32,33]. Song et al. demonstrated that Rv3221a, an anti-sigma factor known as RshA in the same operon, [30] interacts with SigH at a 1:1 ratio [30], leading to SigH inhibition in vitro. Under oxidative stress, phosphorylation of RshA by PknB causes disruption of the RshA and SigH interactions, thereby regulating the induction of the oxidative stress response in mycobacteria [23]. 

### 1.3. Drugs Effective under Latent Stage

Pyrazinamide (PZA) is the only drug that kills MTB in a latent state, which has successfully reduced the time span of TB therapy from nine to six months [34,35,36]. PZA is a prodrug that depends on MTB encoded pyrazinamidase (PZase) (Figure 1A), whose activity is essential for the activation of PZA into the active form, pyrazinoic acid (POA). The POA targets ribosomal protein S1 (RpsA), aspartate decarboxylase (*panD*) (Figure 1D). The earlier protein helps in trans-translation while the latter is involved in ATP synthesis [37,38]. POA binds with RpsA, disrupting the complex of RpsA–tmRNA (Figure 1B). Recently, a large number of PZA-resistance cases have been reported, affecting the latent TB treatment. In our recent study, we evaluated the mechanism behind PZA-resistance in RpsA and PncA, showing a significant effect of mutation in PncA on protein activity [39,40,41,42,43,44,45]. Due to the large number of PZA-resistance cases, the latent stage of TB may function as a reservoir for transmission, affecting the global TB end control program.

### 1.4. Marine Natural Products against Latent TB

With increased resistance to PZA, alternative novel bactericidal against non-replicating MTB will be important to reduce the transmission in the population and also for short period treatment. The screening for alternatives to PZA under latent TB from natural products is a validated approach. Here, nine out of 12 groups of available drugs are naturally derived [46]. Screening of more diverse natural product libraries has incentivized efforts in recent years [47]. The chemical diversity screening may be extended for marine natural products as more diverse and active products have been reported in marine environments [48]. Secondary metabolites that are produced by marine organisms have been found to be effective against many disease causing microorganisms [49]. Recently, some active compounds have been sourced from marine organisms against latent MTB isolates [47,50]. 

In a study by Felix et al. [51], a library of MNPs was screened where four among five compounds (Figure 2) were active against latent MTB isolates, containing 2 puupehenone group metabolites (Table 1). Propane,1,2-diol was not effective against dormant MTB isolates. These dormancy-active hits could reveal novel druggable targets under latent stage, and therefore may lead to an alternative of PZA. 

The core drug regimen has not been modified despite continuous efforts after a prolonged time [52]. Alternatively, research on MTB during in vivo has explored subpopulations of distinct metabolic states within a single host [53]. This knowledge may be useful to uncover the essential activities required for the survival of nonreplicating inhibition [54]. Whole cell screening under the dormancy–activating signaling pathway may provide a direct path to discovering novel bactericidals against latent MTB isolates. Here, the main goal of our study was to highlight the importance of marine drugs [55,56,57] that could effectively kill dormant bacteria as well as the analysis of some signaling pathways for more potent drug target identification.

## 2. Results

A total of 42 articles were retrieved from 1994 to 2018 containing data about the regulations and interactions of genes and proteins under latent MTB isolates where 14 manually searched papers also contained relevant information. 

### 2.1. SigH Regulatory Network

We confirmed the interaction among the SigH regulons where a network file of “string” database in tsv format exposed different paths through Pathlinker of Cytoscape (Appendix A). The stress responder, SigH, has a major role in controlling the response of pathogens to go into latent state and also in the survival of the pathogen. The regulatory pathway is shown in Figure 3A. The external stress signals are sensed by the membrane receptor protein PknB, initiating the signal transduction pathway by phosphorylating the SigH-RshA complex. This phosphorylation causes the disintegration of the SigH-RshA complex, allowing SigH to form a complex with RNA polymerase (RNA-pol), activating a series of stress responders. Upon activation, SigH and RshA are also synthesized, but RshA is continuously inactivated as long as the stress is sensed by PknB. The pathway is negatively regulated by RshA, while positively regulated by phosphorylated RshA (RshA-P), depending on the presence or absence of a stress signal (Figure 3A).

### 2.2. Paths Identification in the Network

All of the stress responders were used as input proteins and the interaction network was searched in a string database. The file (Appendix A) was imported into Cytoscape, where a total of 12 paths were identified using the Pathlinker plugin. The plugins computed multiple paths from the sources to targets where the longest path shown (Appendix A) was found to be most similar to the literature mapped SigH pathway (Figure 3A). The longest path linked all of the essential proteins regulated under the latent stage of MTB. 

### 2.3. SigH Regulation and Marine Drugs

The interacting entities were subjected to six and nine different state stochastic simulations for 100 seconds in the active and inactive states to evaluate the dynamic behavior under latent state of MTB (Figure 3B,C). Stochastic simulations validate the desired functioning of the proposed biological regulatory systems (redox response) as shown in Figure 3. The SigH regulatory pathway may play a crucial role in the survival of the pathogen under extreme stress environment. 

## 3. Discussion

Currently, PZA is the only drug regularly prescribed along with other first-line drugs for the effective control of dormant MTB and is recommended in sensitive as well as multi and extensive drug resistance. However, due to an increased number of PZA-resistance, alternative sources of natural products that are active against the dormant isolate in acidic pH are continuously being sought. Sponges in the marine environment are rich sources of such compounds. Quinones with high selectivity against dormant MTB are from a sponge from the Petrosia (Strongylophora) genus. Terpene quinones including puupehenone metabolites have been extensively studied for their antimicrobial and cytotoxic properties [58,59,60]. The puupehenone derivatives showed anti-TB activity as reported earlier [50]. The MIC of 15-methoxypuupehenol was 20-fold lower and effective against dormant, but not replicating. Puupehedione and Puupehenone had been previously extracted from the sponge Hyrtios sp. [58]. Puupehedione had minor activity against replicating MTB [50], however, a 6-fold selectivity of puupehedione against dormant MTB was still observed.

Puupehenone metabolites inhibit NADH oxidase activity in submitochondrial particles [61,62]. Weinstein et al. observed a bactericidal effect for M. tuberculosis’s type II NADH oxidase (NDH-2) inhibitors in a murine model [63,64]. Inhibitors of NDH-2 proteins such as thioridazine, exhibited bactericidal activity against dormant MTB when compared to replicating isolates [65]. The synthesis of puupehenone enables a path for these molecules [66]. Characterization of the molecular targets for these antimycobacterial marine natural products with selective activity against dormant MTB will be helpful for exploring the insight mechanisms of the survival of dormant MTB under the latent stage of infections. 

The halicyclamine alkaloids (HA) with piperidine rings, haliclonacyclamines A and B (Figure 2) C-1 and C-2 [67], 22-hydroxyhaliclonacyclamine B (C-3) [68], and halicyclamine A (C-4) (Figure 4) [69] have been discovered from the marine sponge Haliclona sp. [70]. Haliclona sp. is one among the dominant sponges at Heron Island, occurring at depths of about 15 m in reef slope [71]. The anti-TB bactericidal and bacteriostatic activity of C-3 and C-4 were evaluated under aerobic and hypoxic conditions. The colony forming unit (CFUs) of M. bovis BCG was not detected after eight days and 10 days of incubation under aerobic and hypoxic environment, respectively, indicating bactericidal activity of C-2. Compound C-4 exhibited a strong cidal effect against mycobacterium sp. including M. tuberculosis H37Ra (MICs of 2.16–10.82 µM) under dormant state. Compounds C-1 and C-2 also exhibited cidal activities against replicating and non-replicating (latent state) of MTB. C-3 exhibited weak activity that might be due to the 22-hydroxy group. Compound C-4, which is involved in catalytic conversion of inosine monophosphate to xanthosine monophosphate in the de novo synthesis of guanine nucleotides, was isolated as an inhibitor of inosine 5′-monophosphate dehydrogenase (IMPDH) [69]. IMPDH was cloned into M. smegmatis to study the anti-TB mechanism. However, both the wild-type M. smegmatis and IMPDH over-expressing strains exhibited similar MIC values, indicating that IMPDH was not the target of compound C-4 [70]. Another HA, neopetrosiamine A (C-5) (Figure 2), isolated from sponge Neopetrosia proxima growing near Puerto Rico in a marine environment [72], showed good cidal activity against MTB H37Rv (MIC:17.05 µM).

The stress response under the latent state of MTB is mediated by many regulatory genes and the role of SigH in the oxidative stress was first established by Fernandes et al. [29] through experiments using *M. smegmatis* SigH mutants. SigH is a major MTB regulator that provides protection from reactive oxygen species generated by the human host [31,73]. The SigH-encoded protein protects MTB against oxidative stress by regulating the expression of the stress-responsive factors SigE and thioredoxins trxB1 and trxC. The stress-responsive “S” factor and SigB were also regulated by the SigE and SigH regulators. However, the mechanism of SigH regulation was not clearly explored, and neither were there any synthetic biology approaches applied for better understanding. Marine compounds shown in Figure 2 may be applied against dormant isolates to find their effect. Furthermore, the mechanism may also be through a knockout system.

Prokaryotic RNA-pol may be a potent target as it plays a role in the initiation of the complex network. RNA-pol is directed by sigma factors toward specific promotors through the formation of a sigma/RNA-pol holoenzyme, which may be a closed stable complex that is ineffectual for transcription initiation (sigma 54), or may proceed directly to an open complex that is capable of transcription (sigma 70). RNA-pol is an ideal drug target for a number of antibiotics because it is an integral part of a crucial cellular process [74,75].

## 4. Methods

### 4.1. Literature Search

To map a signaling pathway under the latent stage of MTB, the RISmed package of R was used to retrieve relevant literature from the Entrez Utilities to the PubMed database at National Center For Biotechnology Information (NCBI) [76,77,78]. The RISmed package is fast and time efficient, extracting the exact information. The key words, latent stage TB, dormant state MTB, MTB survival under stress, role of sigma factors, SigH of MTB, regulation of MTB pathway under stress condition, RshA role, role of TrxC, PknB, and stress regulation were used to mine the relevant information from the literature databases. Subsequently, all the relevant papers were manually searched for the genes and proteins expressed under the latent stage. 

### 4.2. Pathway Construction Using Systems Biology Approach

All the major regulators involved in signaling and interactions were extracted from the literature and mapped in the form of a regulatory network in UPPAAL, an integrated tool for modeling [79], in the form of a pathway.

### 4.3. Validation of SigH Regulatory Pathway

The SigH regulatory pathway mapped from the literature was further confirmed for their interactions in a “STRING” database [80]. All the SigH regulons were entered as input in the string database to observe the interacting network. The protein network was further increased by the addition of more nodes (proteins) until all of the extracted entities were found to be interconnected in a single network. The network file was downloaded in the Tab Separated Values (TSV) format (Appendix A) and imported into Cytoscape v 3.5.1 [81] where different paths were generated inside the network using the Pathlinker [82] plugin in Cytoscape. The sources and targets were selected based on the mined literature data, and the longest path was searched using a background protein interaction network. Pathlinker requires three inputs: a (directed) network G, a set S of “sources”, and a set T of “targets”. Each element of S and T must be a node in G. Pathlinker efficiently computes several short paths from the receptors to transcriptional regulators (TRs) in a network and can accurately rebuild an inclusive set of signaling pathways from the NetPath and KEGG databases. Pathlinker has a higher precision and recall when compared to several state-of-the-art algorithms. The longest path was analyzed based on the score of Pathlinker using the ANIMO plugin [83] for visualization purposes.

### 4.4. Synthetic Biology and SigH Activation 

The pathway was simulated for 100 s using the Java Script [84] to analyze the effect on the active and inactive state of SigH regulation. 

## 5. Conclusions

The marine environment, a highly valuable source for new lead structures, is a rich source of anti-TB bioactive compounds that have the potential to be used as an alternative to PZA. The biological activity of these leads gives hope for effective anti-TB agents that will show low-toxicity under the latent stage. Here in this review, we highlighted some marine natural products that are effective against latent TB. Furthermore, we mapped a novel SigH regulatory pathway whose regulons may be patent targets to verify the mechanism of action. Although studies have been carried out to discover these agents, the mechanism of action is still uncertain and will require future research. These compounds may be tested against the potent targets of the SigH signaling pathway, which is required for the survival of MTB under different kinds of stress including oxidative and acidic stress. This study provides useful information about the screening of marine natural products active against latent TB that may be tested against the signaling pathway under the latent stage of MTB. 

## Figures and Tables

**Figure 1 marinedrugs-17-00549-f001:**
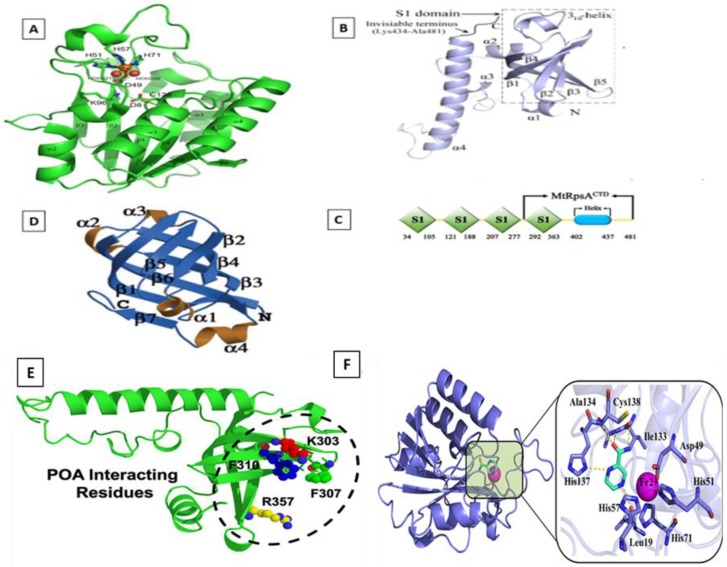
Crystal structures of PZA and POA targets. (**A**) PZase; (**B**) RpsA; (**C**) Domain organization of RpsA and its C-terminal domain (MtRpsACTD); (**D**) PanD. PZase converts PZA into POA inhibiting the activity of PanD and RpsA. POA interactions with RpsA (**E**) and PZA with PZase (**F**). The prodrug PZA is converted into active form, POA, inhibiting the trans-translational proteins (RpsA).

**Figure 2 marinedrugs-17-00549-f002:**
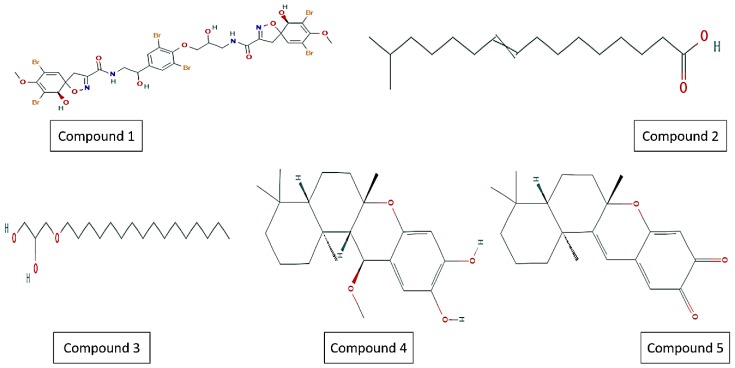
Compounds active against latent TB (dormant state). Compound 1, fistularin-3/11-epi-fistularin-3; compound 2, 15-methyl-9(*Z*)-hexadecenoic acid; compound 3, (hexadecyloxy) propane,1,2-diol; compound 4, 15- alpha methoxypuupehenol; and compound 5, puupehedione. Compound 3 exhibited mild activity against replicating MTB (active TB).

**Figure 3 marinedrugs-17-00549-f003:**
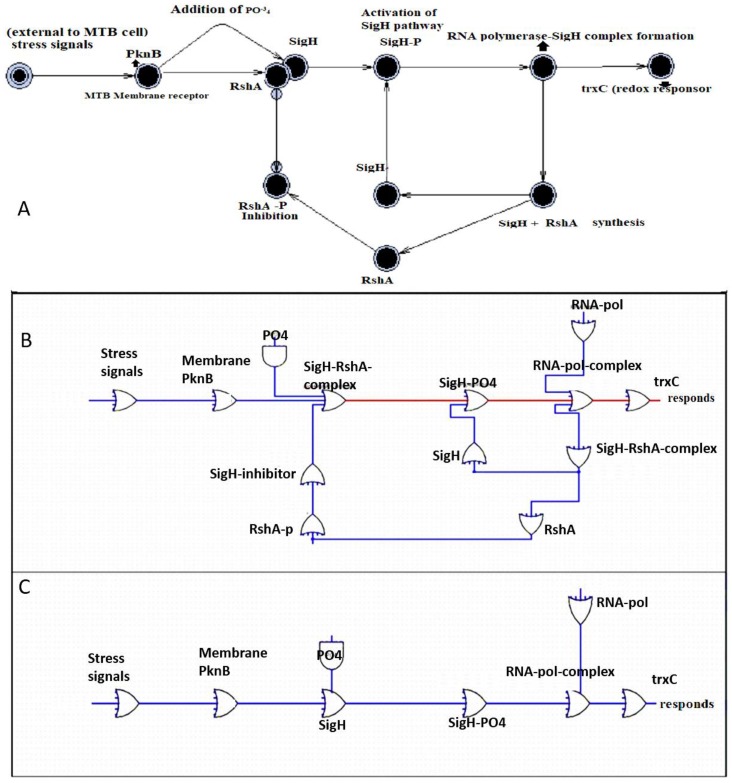
SigH signaling pathway under stress conditions. (**A**) The SigH signaling mechanism. (**B**) Boolean network simulation with the SigH-RshA complex negatively regulated the pathway, deactivating the regulatory pathway. (**C**) Boolean network simulation without the SigH-RshA complex, where the “SigH-PO_4_” activates the mechanism. RshA-P: phosphorylated RshA; SigH-PO_4_: phosphorylated SigH.

**Figure 4 marinedrugs-17-00549-f004:**
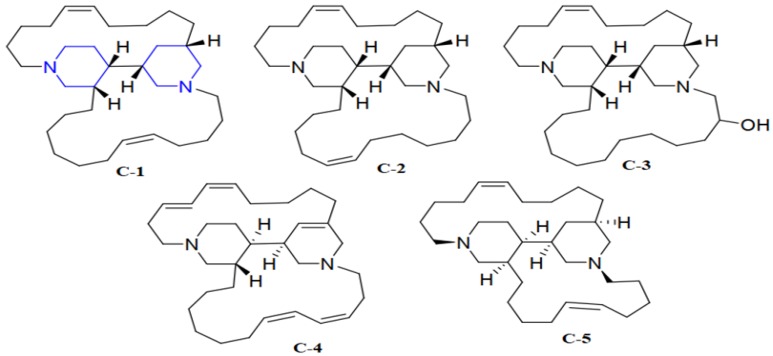
Anti-TB compounds, C-1 to C-5 active against latent TB. The blue colored lines were the same in all compounds. Haliclonacyclamines A and B (C-1 and C-2). 22-hydroxyhaliclonacyclamine B (C-3), Halicyclamine A (C-4), and neopetrosiamine A (C-5).

**Table 1 marinedrugs-17-00549-t001:** Biological profile of marine pure compounds against dormant MTB isolates adopted from Felix et al; 2017 with permission from 2017 American Society for Microbiology [51].

Compound	Formula	Molecular Mass (kDa)	MIC_R_ (g/mL) ^a^	MIC_D_ (g/mL) ^b^	MIC_R_/MIC_D_	IC_50_ (g/mL)	SIR ^c^	SID ^d^	Source
1	C31H30Br6N4O11	1,114.02	8.5	Inactive	NA	200	23.5	NA	HBOI.047.F07
2	C19H40O3	316.53	60.8	22.5	2.7	200	3.3	8.5	HBOI.047.F07
3	C16H30O2	254.41	28.5	7.9	3.6	200	7.0	31.1	HBOI.031.C02
4	C22H32O4	360.49	11.3	0.5	21.8	8	0.7	15.5	HBOI.050.F04
5	C21H26O3	326.44	87.6	15.4	5.6	50.4	0.6	6.2	HBOI.050.F04

^a^ MIC_R_, MIC against replicating Mtb-Lux. ^b^ MIC_D_, MIC against dormant Mtb-Lux. ^c^ SIR, SI for replicating Mtb-Lux. ^d^ SID, SI for dormant Mtb-Lux.

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
