# Peer review of "Marine Natural Products and Drug Resistance in Latent Tuberculosis"

_marinedrugs, 2019, doi:10.3390/md17100549_

Round 1

Reviewer 1 Report

The authors conducted an important analysis.  This is an interesting paper analyzing the major regulatory 30 pathway under latent TB that might be used for screening of selective inhibitors among marine 31 natural products (MNPs). This manuscript is very well developed, the results contextualization and the manuscript is well written. The data presented are new and relevant

Author Response

Dear reviewer Thanks for your kind comments

Reviewer 2 Report

Dear Chief of Editor-

Sorry for late reply.

I need to discusse this manuscript with the structure researcher in our team. 

I am focused on the logic and novelty of this manuscript, but I did not feel that I had found important and novel findings. As comments from the structure guy, he also agree that the quality and analysis of this research was not very high.

In addition, english is not very fluent. 

Ho Seong Seo

Author Response

Dear reviewer thanks for your kind comments.

In this paper focused on latent TB which a major source of transmission in high burden countries including Pakistan and China.

At the moment, PZA is the only drug recommended by WHO for treatment of latent (dormant) isolates.

Due PZA resistance and also the importance and diversity of marine drugs, We extracted marine natural products acting best against dormantmycobacterium tuberculosis.

To find the potent targets under latent state, We searched literature containing genes/proteins interacting under dormant stage. All these proteins were mapped i the form regulatory pathway. 

The pathway regulatory behavior and dynamic was evaluated through simulation.

We found that targeting the pathway proteins may causes the inhibition of MTB growth under dormant  stage. 

Marine drug interactions against these target may be confirmed after simulation or experimental approach.

The study provide useful information about the potent marine natural product extracted from literature and their targets i.e. major regulatory pathway that may be a useful target as alternative of PZA. 

English language have redefined through a friend, native english speaker known as kerick in california. 

Reviewer 3 Report

Manuscript Number: marinedrugs-585794

entitled: Marine Natural Products and Drug Resistance in Latent Tuberculosis

It’s an interesting paper. I have the following question/comments to the authors.

Abstract;

Some part of abstract should be moved to introduction part ”Abstract: According to the World Health Organization report 2018, 1.7 billion people (23% of the world’s population) are estimated to have a latent TB infection, indicating a risk of developing active TB during their lifetime.” In this part must be ONLY the most important achievements, subjects and for rev. is important to say what period of time is checking e.g. until June 2019.

Keywords: please use more specific keywords than Marine drugs,

For broad readerships will be good to draw structures of the most important drugs like “pyrazinamide (PZA) is the only drug”. Many readers only quickly watch papers.

Figure 2. Please use only one style to draw molecules.

Line 135 “A total of 42 articles” please add information about period of time.

Figure 4. There is no information in the text why part of C-1 molecule is in blue.

In my judgment, there is good publishable science in this manuscript, but it needs some work before it can be accepted.

Author Response

Comments and Suggestions for Authors

Manuscript Number: marinedrugs-585794

entitled: Marine Natural Products and Drug Resistance in Latent Tuberculosis

It’s an interesting paper. I have the following question/comments to the authors.

Abstract;

Some part of abstract should be moved to introduction part ”Abstract: According to the World Health Organization report 2018, 1.7 billion people (23% of the world’s population) are estimated to have a latent TB infection, indicating a risk of developing active TB during their lifetime.” In this part must be ONLY the most important achievements, subjects and for rev. is important to say what period of time is checking e.g. until June 2019.

Author response: We appreciate the reviewer suggestions. The abstract was corrected as per reviewer comments.

Keywords: please use more specific keywords than Marine drugs,

Author response: We appreciate the reviewer suggestions. Keywords corrected as per reviewer comments.

For broad readerships will be good to draw structures of the most important drugs like “pyrazinamide (PZA) is the only drug”. Many readers only quickly watch papers.

Author response: We appreciate the reviewer suggestions. We did as per reviewer suggestion please.

Figure 2. Please use only one style to draw molecules.

Author response: We appreciate the reviewer suggestions. We did as per reviewer suggestion please.

Line 135 “A total of 42 articles” please add information about period of time.

Author response: We appreciate the reviewer suggestions. We did as per reviewer suggestion please.

Figure 4. There is no information in the text why part of C-1 molecule is in blue.

Author response: We appreciate the reviewer suggestions. Information was added as per reviewer suggestion please.

Regards

Authors

Round 2

Reviewer 2 Report

I reviewed this paper as a research manuscript, since there are the sections containing Result and Discussion. However, the second round of review, I found out that this is "Review" paper. I want to politely apologize for this.

However, my main concern is the format. It would be nice to have them organized in the form of a review article. 

Author Response

Thanks for your kind comments. We apreciate the reviewer comments. 

We appreciate your concern but as it is a review article and we have adopted a certain methodology to explore a pathway for future marine drugs experiments on our pathway. Therefore, as per journal style requirement, methodology was kept at the last please.